# Empowering Parents of Adolescents at Elevated Risk of Suicide: Co-Designing an Adaptation to a Coach-Assisted, Digital Parenting Intervention

**DOI:** 10.3390/ejihpe15100199

**Published:** 2025-09-29

**Authors:** Alice Cao, Ling Wu, Glenn Melvin, Mairead Cardamone-Breen, Grace Broomfield, Joshua Seguin, Chloe Salvaris, Jue Xie, Dhruv Basur, Tom Bartindale, Roisin McNaney, Patrick Olivier, Marie Bee Hui Yap

**Affiliations:** 1School of Psychological Sciences, Turner Institute for Brain and Mental Health, Monash University, Clayton 3800, Australia; alice.cao@monash.edu (A.C.); mairead.cardamone-breen@monash.edu (M.C.-B.);; 2Action Lab, Faculty of Information Technology, Monash University, Clayton 3800, Australia; ling.wu@monash.edu (L.W.); joshua.seguin@monash.edu (J.S.); jue.xie@monash.edu (J.X.); dhruv.basur@monash.edu (D.B.); patrick.olivier@monash.edu (P.O.); 3SEED Lifespan, School of Psychology, Deakin University, Burwood 3125, Australia; 4Department of Computer and Information Sciences, Northumbria University, Newcastle upon Tyne NE1 8ST, UK; tom.bartindale@northumbria.ac.uk; 5School of Computing and Information Systems, University of Melbourne, Parkville 3010, Australia; roisin.mcnaney@unimelb.edu.au; 6Melbourne School of Population and Global Health, University of Melbourne, Parkville 3010, Australia

**Keywords:** youth, parent, empowerment, online, suicide prevention, lived experience, co-design, technology, intervention

## Abstract

Suicidal ideation and behaviours are common among adolescents. Parents play a fundamental protective role in the prevention of adolescent suicide, but many describe feeling ill-equipped in their caretaking role. This is despite prior research indicating that it is important for these parents to feel empowered to emotionally support their adolescent if they are experiencing suicidality. An online parenting program could offer parents flexible access to evidence-based parenting strategies. However, there are limited digital resources for these parents and, further, very little is known about how an intervention could be designed to support the empowerment of these parents. Therefore, the aim of the current study is to explore how an existing evidence-based, digital parenting intervention, Partners in Parenting (PiP+), could be adapted through co-design to empower parents. Four parents who have lived experience of caring for a suicidal adolescent, four young people who experienced suicidality during adolescence, and four experts in youth mental health/suicide prevention participated in four sets of co-design workshops to innovate adaptations to PiP+ to empower parents of suicidal adolescents. Affinity mapping was used to analyse and interpret findings. Three key themes highlight how a digital intervention could be innovated and adapted to empower parents caring for a suicidal adolescent. Specifically, for parents to feel empowered to parent a suicidal adolescent, a digital intervention should support them to (1) “deal with the now”; (2) “acknowledge needs and understand their role”, and (3) “hold hope for the future”. Further, ten sub-themes were developed illustrating different concepts related to these themes. Findings highlight how technological features could support parents to feel more empowered when caring for a suicidal adolescent. In conclusion, the proposed technological features illustrate how digital interventions can be adapted to empower parents in their role of emotionally supporting and managing the suicide risk of their adolescent.

## 1. Introduction

Anxiety disorders and depression (also known as internalising disorders) are at peak prevalence and onset during adolescence. Concerningly, adolescents with internalising disorders are 1.54 times more likely to suicide ([19]), and suicide is the fourth leading cause of death among adolescents worldwide ([32]). Beyond the devastating loss of life, many more adolescents experience suicidal thoughts and behaviours ([18]). Yet despite the distress of these adolescents, research has found that the majority of these adolescents (approximately 50% to 60%) do not seek professional help ([11]; [16]). Rather, adolescents with mental disorders are more likely to seek support from informal sources, and, in particular, their parents ([34]). 

Parents present as opportune partners in adolescent suicide prevention. A systematic review identified that parental factors such as warmth, autonomy granting, and behaviour control are negatively associated with adolescent suicidality and internalising disorders ([14]). Further, adolescents at risk of suicide are typically cared for at home ([10]) and parents play an important logistical role in restricting adolescents’ means of suicide, risk monitoring, and providing emotional support ([10]). Prior research also highlights the importance of parental self-efficacy (to engage with adolescent suicide prevention strategies), showing that higher parental self-efficacy is associated with fewer adolescent suicide-related emergency department admissions at a four-month follow-up ([10]). 

Yet, despite the critical role of parents, most parents caring for a suicidal adolescent report feeling overwhelmed and ill-equipped to support their adolescent experiencing suicidality ([15]; [26]). In line with parents’ reports, there is currently no digital parenting intervention that is available to support parents who are caring for a suicidal adolescent. Given that most parents prefer receiving general parenting information through online platforms as opposed to the more traditional face-to-face approach, the lack of digital parenting interventions for adolescent suicidality is particularly concerning ([4]; [22]). In particular, digital interventions have the potential to help overcome logistical barriers that parents commonly report facing, including scheduling conflicts, transportation, and other accessibility issues ([27]). 

While the literature consistently highlights calls for greater parental support, specifically for parents of suicidal adolescents ([7]; [21]; [26]), very little is known about parents’ support needs when caring for a suicidal adolescent. Only one study, to date, has considered parents’ support needs in the context of developing an online parenting intervention for parents of suicidal adolescents; a key finding of this study was that parents caring for suicidal adolescents needed to feel empowered ([8]). Parent empowerment was conceptualised as the belief that they have a valuable role in preventing their adolescent’s suicide and undertaking the behaviours to do so ([8]). Further, the field of human–computer interaction has highlighted empowerment as a key factor in designing effective interventions ([29]). Yet, despite research calling for digital interventions to ‘empower’ its end users, the term ‘empowerment’ remains elusive and challenging to characterise and quantify ([29]). To date, little is known about how a technological system could be designed to promote parent empowerment in the context of caring for a suicidal adolescent. Co-design presents as an ideal method to consider how a technological system could be designed to better meet the needs of these parents. Co-design is a participatory research method that aims to engage its intended users and key stakeholders in the planning, design, and evaluation process, thus facilitating a better understanding and meeting the needs of its end users ([6]; [30]).

Given the overlap in parental factors associated with adolescent internalising disorders and suicidality, a parenting intervention that addresses both appears as a promising path forward. Currently, there is no purely digital parenting intervention that exists for parents of suicidal adolescents, although a digital intervention does exist for parents of adolescents with internalising disorders which has pre-established acceptability, feasibility, and usefulness ([12]). Thus, leveraging an existing parenting program with pre-established acceptability, feasibility, and usefulness for the parent and their adolescent may expedite the development process. 

One such parenting intervention is the Partners in Parenting (PiP) program, which aims to reduce the risk and impact of adolescent depression and anxiety disorders by equipping parents with evidence-based parenting strategies. The Partners in Parenting program is a multi-level, individually tailored, web-based parenting intervention, spanning universal prevention (Level 1) to early intervention for clinical-levels of adolescent internalising problems (Level 4) (see [33] for more details). The highest level (Level 4; henceforth referred to as PiP+) combines the online self-directed program with regular one-on-one coaching sessions via video conference ([12]). PiP+ involves up to nine interactive online modules covering different domains of evidence-based parenting ([12]), and up to 13 video-conferencing sessions with their therapist-coach, occurring weekly or fortnightly ([12]). Although PiP+ targets parenting behaviours with adolescent internalising disorders (e.g., parenting risk and protective factors), some of which are also associated with reducing adolescent suicide risk, PiP+ is not designed for parents of suicidal adolescents.

To best understand how parents can be empowered in their role of caring for a suicidal adolescent requires recognising the broader context in which these experiences unfold. As such, a digital intervention that aims to empower parents should consider the lived experience of these parents through a multifaceted lens, encompassing the viewpoints of the parent, adolescent, and healthcare systems supporting the adolescent’s mental health ([20]). It has been argued that recognising the diversity of perspectives from different key stakeholders can increase the likelihood that interventions will have more meaningful outcomes ([20]).

The study aims to explore the empowerment of parents caring for a suicidal adolescent through a qualitative design. A secondary aim is to explore how parental empowerment can be embedded within a technological system (i.e., PiP+ through both self-directed online modules and digitally mediated coaching sessions).

## 2. Materials and Methods

### 2.1. Study Design

This qualitative study used a co-design approach ([30]; [31]) to adapt an existing digital parenting intervention (PiP+; [33]) for parents of adolescents experiencing suicidality (PiP-SP+). We adopted an inductive approach to identify themes which would facilitate the empowerment of parents caring for a suicidal adolescent, and then a deductive approach to validate whether the digital features suggested from such themes would support parents to feel empowered when caring for their suicidal adolescent. Three stakeholder groups were engaged, (1) parents, (2) young people, and (3) experts in youth mental health/suicide prevention, across four sets of online workshops. The first set of workshops was with parents, and was designed to elicit their perspectives on the enablers and barriers to empowerment. The second set of workshops was with young people, and was designed to capture their views on the acceptability of parent empowerment strategies. The third set of workshops, conducted with professional experts, was designed to gather their input regarding the feasibility of adapting a therapist-assisted online parenting program (PiP+), exploring systemic factors to empowerment. The final set of workshops was a sense-checking workshop with parents, designed to validate and refine themes and digital intervention features with parents.

### 2.2. Ethical Approval

Prior to commencing recruitment and any engagement with participants, ethical approval was sought and granted by Monash University Human Research Ethics Committee (ID #28055). As the study was open for young people who were under the age of 18 to participate, both the parent and the young person were asked to review the explanatory statement written for young people under the age of 18. If both agree, the parent must provide informed consent. Therefore, prior to participation, written informed consent was obtained from all participants. 

### 2.3. Recruitment and Participants

Participants were recruited by sharing flyers with professional networks, online community noticeboards, and social media sites. Parents, young people, and experts who participated in a previous research study on understanding the lived experience of parenting a suicidal adolescent ([8]) and who provided consent to be contacted for future research were also invited. 

All participants needed to live in Australia, speak English and have stable internet access. Parent participants must have lived experience of parenting an adolescent (aged 12–18) who experienced suicidal thoughts or behaviours. Young people (aged 15–25) were eligible if they had lived experience of suicidal thoughts or behaviours when they were adolescents (aged 12–18). Lastly, professionals were eligible if they had over 3 years of experience working in the field of youth mental health and suicide prevention in either clinical or research roles (henceforth referred to as experts). Each participant was reimbursed $40 AUD per hour. Notably, to the best of the researcher’s knowledge, no stakeholders held additional relationships with one another. 

### 2.4. Data Collection

The research team developed four co-design workshops and respective activities specific to each participant group, presented below in Section 2.4.1, Section 2.4.2, Section 2.4.3 and Section 2.4.4. Workshops were held in November and December 2021, each lasted 1 to 2.5 h, were conducted using Zoom videoconferencing software version 5.8.3, and used screen share to show participants a Google Slides deck. The facilitator used text boxes to annotate the slides with participant insights in real time.

#### 2.4.1. First Co-Design Workshop with Parents: Enablers and Barriers to Parental Empowerment

Two workshops were conducted—one group workshop with three parents, and an individual workshop due to logistical constraints. This first set of co-design workshops aimed to understand how an intervention could be designed to enable empowerment at different stages of supporting a suicidal adolescent, exploring potential enablers and barriers to empowerment. As parents with lived experience of supporting a suicidal adolescent commonly report feeling helplessness, guilt, and self-blame ([15]), we intentionally focused the workshop on a vignette of a parent who was seeking the parent participants’ lived experience advice to support their suicidal adolescent. Vignettes can encourage parents to afford themselves the same kindness they would to a friend and provide a third-party perspective, which is often less self-critical ([13]).

The co-design activity asked parents to create a “magic machine” that would empower their hypothetical friend who was caring for a suicidal adolescent (see Figure 1), thus exploring the enablers of parent empowerment. The magic machine activity encourages participants to be imaginative and limitless in their design suggestions ([2]; [3]). Ultimately, this approach aimed to help parents draw from a position of empowerment to problem-solve rather than a position of helplessness, guilt, and distress. These workshops also explored emotional enablers and barriers to parental empowerment by asking parents to indicate what emotions they thought their hypothetical friend would be feeling at different stages of supporting their suicidal adolescent. Figure 1 presents exemplar slides from the first parent workshop, showing the magic machine exercise and the emotion-mapping task to explore parental enablers and barriers to empowerment.

#### 2.4.2. Co-Design Workshop with Young People: The Acceptability of Parent Empowerment in the Context of Suicide Prevention

After the initial parent workshops, four separate individual co-design workshops with adolescent stakeholders were conducted. These workshops were conducted separately as all young people but one indicated that their preference was to engage individually. The focus of these workshops was to explore what would be the most helpful and acceptable ways for parents to support their adolescents experiencing suicidal thoughts or behaviours. 

Similarly to the parent workshops described in Section 2.4.1 we wanted young people to draw their lived experience from a place of empowerment, as opposed to distress and helplessness. Thus, we provided young people with a vignette and asked them to imagine being a radio host who had overcome their mental health challenges and were now providing advice to an imaginary parent experiencing difficulties in supporting their adolescent facing suicidality. We asked young people to provide advice on how parents could build a communicative relationship with their adolescents while monitoring for signs of suicidality in ways that would be deemed acceptable. 

#### 2.4.3. Co-Design Workshop with Experts: Feasibility of a Therapist-Assisted Online Parenting Program to Support Parent Empowerment

Following workshops with young people, a group co-design workshop was conducted with four experts who were all clinicians and researchers in youth mental health or suicide prevention. The focus of this workshop was to explore how digital, therapist-assisted parenting interventions could be adapted to empower parents while considering systemic factors (e.g., how parents can be supported to navigate the healthcare systems, school systems, and broader family dynamics, including siblings). Experts were considered well-placed to provide suggestions on navigating the aforementioned systems in ways that would empower parents.

As such, experts were provided with the context of the PiP+ program which described what components parents would receive; specifically, online modules and coaching sessions. Experts were then presented with four vignettes of coaching sessions transcriptions between a parent and coach, and were asked how they would supervise the PiP+ coach whose aim is to empower parents of suicidal adolescents when facing these systemic barriers. Vignettes explore systemic barriers such as how parents can monitor their adolescent’s safety while maintaining the relationship with the adolescent, supporting parents to find their own mental health supports, if and how parents should communicate with the adolescent’s care team (e.g., psychologists, teachers), and how to navigate changes in the family dynamic with siblings when one adolescent is experiencing suicidal thoughts or behaviours. Consequently, the facilitator asked experts questions such as “what constructive feedback would you give the PiP+ coach to support this dilemma?” and “How would you have dealt with this scenario instead?”. Figure 2 presents an exemplar slide in the expert workshop, depicting a vignette of a parent-coach conversation for experts to discuss strategies which could help overcome systemic barriers.

#### 2.4.4. Final Co-Design Workshop with Parents: Sense-Checking Themes of Parental Empowerment

We invited all four parents from the first set of parent co-design workshops to participate in the final parent workshop; three parents agreed to participate. The final participants of this group co-design workshop were mothers aged 39 to 52 (M = 47.00, SD = 5.72).

The final co-design workshop aimed to sense-check the identified sub-themes triangulated from all sets of workshops conducted previously by exploring whether the identified sub-themes were reflective of parents’ lived experiences and if they were considered important to parent empowerment when caring for a suicidal adolescent. Further, the research team had extrapolated digital design solutions that may address the needs identified by themes developed from parents, young people, and experts in the prior workshops. The lead author brainstormed potential features which could align with the sub-theme, and the concept of these features were then iteratively refined through multidisciplinary team discussions including human–computer interaction researchers, psychology researchers, and psychologists regarding how a feature could align with the needs and preferences articulated by stakeholder groups. These identified sub-themes and extrapolated features in a digital, therapist-assisted, parenting intervention are presented in Table 1. For example, the identified sub-theme of ‘self-efficacy’ was conceptualised as a digital feature of a personalised plan that parents can use to prevent their adolescent’s suicide.

We listed each feature on a virtual “post-it” note, presented in Figure 3. Parents were asked to sort the features based on importance and usability by placing the notes on a two-dimensional axis (axis 1: most important to least important; axis 2: most usable to least usable). During the workshop, we asked parents if and how the feature mapped onto the identified theme (e.g., “what does this feature mean for a parent’s feelings of self-efficacy?”). Additionally, we asked specific follow-up questions to consider how each feature could be implemented (e.g., “how would this feature work best?”). Figure 3 presents a slide from the final parent workshop, where parents sorted the digital features of a personalised action plan along axes of importance and usability.

### 2.5. Data Analysis

All interviews were initially transcribed by a trusted third-party Artificial Intelligence software program, Descript version 7.0.4 for Mac (Descript, San Francisco, CA, USA; see https://www.descript.com/). A.C. manually corrected the interviews for accurate verbatim transcription. Data was analysed using affinity mapping, a co-design analytic technique which involves visually clustering qualitative data to identify patterns and relationships ([5]; [17]; [23]; [24]). The first author AC and author GB familiarised and read through the transcripts and distilled participant insights into key points which were reduced to post-it notes in preparation for affinity mapping. Using the process of affinity mapping, concepts that were considered related to each other were grouped by AC and GB to form candidate sub-themes for each stakeholder group. AC and GB collaboratively integrated data across all workshops to identify candidate themes and sub-themes for empowering parents in supporting a suicidal adolescent in acceptable and feasible ways. The candidate sub-theme clusters were then grouped and further refined into overarching themes. Together, the overarching candidate themes and sub-themes were organised into a thematic map. All candidate themes and sub-themes were presented to the research team and iteratively discussed until the final themes and sub-themes were agreed upon.

## 3. Results

### 3.1. Participant Characteristics

A total of four parents, four young people, and four experts participated in co-design workshops. Table 2 shows the demographic characteristics of the participants.

### 3.2. Identified Themes Facilitating the Empowerment of Parents Caring for a Suicidal Adolescent

Three overarching themes were developed upon triangulating perspectives of parents, young people and experts: 1. Dealing with the now, 2. Acknowledging needs and understanding their role, and 3. Holding hope for the future. These overarching themes represent the commonalities across all stakeholder groups. Additionally, there were 10 sub-themes identified, which reflect unique insights specific to each stakeholder group. Some sub-themes intersect and overlap with other themes, reflecting the interconnected nature of the findings. Figure 4 presents a thematic map to visually represent the relationship between the themes and sub-themes. The thematic map (Figure 4) presents a summary of the overarching themes and sub-themes of parental empowerment when emotionally supporting their suicidal adolescent. Themes are presented in black text, whereas sub-themes identified by each participant group are indicated using a colour legend (orange for parents, blue for young people, and green for experts). Areas of overlap between sub-themes reflect conceptual connections across stakeholder perspectives, to illustrate both shared and unique contributions.

The themes identified which facilitate parental empowerment when caring for a suicidal adolescent are described below, where a description of each theme is provided.

#### 3.2.1. Dealing with the Now

For parents to feel empowered, stakeholders identified that parents need to acquire and draw upon specific skills and knowledge during a suicidal crisis (the acute state which precedes a suicide attempt; [25]). Five sub-themes were identified in the overarching ‘dealing with the now’ theme and are described with illustrative quotes in Table 3. These sub-themes include skills, knowledge, or beliefs that empower parents to support their suicidal adolescent in the moment of a crisis.

#### 3.2.2. Acknowledging Needs and Understanding Their Role

All three stakeholder groups identified the need for parents to acknowledge their needs and roles to foster parental empowerment. Young people and experts emphasised the importance of parents acknowledging their unique role within the adolescent’s care team (e.g., that parents must coordinate the care for their adolescents but are not their health professionals) and prioritising self-care. Parents acknowledged the value of engaging in self-care, having adequate personal support (e.g., from friends and family), and having access to professional support systems. The four sub-themes of ‘acknowledging needs and understanding their role’, the description of the sub-themes, and indicative verbatim quotes are presented in Table 3.

#### 3.2.3. Holding Hope for the Future

Lastly, all three stakeholder groups identified that to foster empowerment within parents, there must be hope that the adolescent’s suicidal thoughts and behaviours can be overcome. If parents can hold hope for their adolescent’s recovery, stakeholders believe that parents are more likely to be empowered in their role of caring for their suicidal adolescent. Three sub-themes of ‘holding hope for the future’ were identified and are described with illustrative quotes in Table 3.

### 3.3. How Empowerment Could Be Embedded in the Technological System

The results of the sense-checking workshop are presented herein. The final workshop aimed to sense-check (validate) sub-themes generated across all prior parent, young people, and expert workshops and to elicit feedback on the preliminary design features mapped to those themes (see Section 2.5 for and Table 2 for the digital features). 

The proposed digital features (see Table 2) were mapped onto a virtual two-axis board (see Figure 5 for results of the activity). The features rated as both important and highly usable included a personalised safety planning tool (mapped to self-efficacy sub-theme), the coach exploring with parent the signs of suicide for their adolescent (understanding adolescent’s suicidality sub-theme), and reflection activity led by coach: parents reflect on how the content applies to a current challenge and their own emotions (mapped to building skills sub-theme). Notably, no features were described by parents as ones that will not be used often and are not important, thus affirming the relevance of the co-designed sub-themes and features.

## 4. Discussion

This study illustrated the facilitators to parental empowerment in the context of caring for a suicidal adolescent by triangulating the perspectives of parents, young people, and experts. Following these findings, features which could be integrated to adapt a digital, therapist-assisted, parenting intervention were extrapolated. Our co-design workshops revealed that to empower parents of adolescents at risk of suicide, the design of an intervention should address three key themes: (1) “Dealing with the now,” (2) “Acknowledging needs and understanding their role,” and (3) “Holding hope for the future”. To operationalise these themes, we propose specific design adaptations to an existing parenting program (PiP+) by leveraging technology-mediated support (i.e., parent-to-coach and parent-to-parenting program interactions), see Table 2 for list of specific adaptations. 

Through the ‘dealing with the now’ theme, it was emphasised that for parents to feel empowered, they needed to acquire and draw upon specific skills and knowledge during a suicidal crisis. Yet to do so, they must first understand their adolescent’s suicidality, develop specific skills, have the confidence to use these skills, and approach their adolescent with an open and trusting attitude. Further, it was emphasised that to ‘deal with the now’, the skills and knowledge parents acquire must be tailored to the unique circumstances of their adolescent. PiP+ would therefore need technological adaptations which would support parents to develop suicide prevention-specific skills. Such adaptations could involve supporting parents to personalise for themselves a digital action plan that they can refer to when their adolescent is suicidal; provide digital module content to understand the signs of adolescent suicidality, or adapt coaching sessions to focus on a parent’s emotional distress associated with caring for a suicidal adolescent and how this can be managed. As such, to ‘deal with the now’, there is a need for highly personalised support which lends itself well to an intervention such as PiP+ given that it combines the online self-directed program presenting evidence-based, actionable strategies with regular coaching sessions via video conference. Following from our findings, it seems unlikely that an untailored parenting intervention without coaches (i.e., a fully self-guided online intervention) would adequately meet the needs of this parent group. The digital features we suggest require a human element, such as a coach, to support parents with the tailoring of knowledge and skill implementation. 

The ‘acknowledging needs and understanding their role’ theme highlighted that to empower parents, they should not feel alone in the care of their adolescent and that parents should reflect upon their unique role within the context of the adolescent’s care team. Prior research has demonstrated that parents often engage with their children in multiple roles and need to shift their role depending on their child’s ever-changing needs; thus, technology should support this process ([28]). Indeed, some guidelines has even been developed to support parents caring for children in the context of their child experiencing trauma to switch between the role of peer, supporter, and carer in technological interventions ([1]). However, our findings challenge these principles, as continual role-switching can lead to feelings of isolation, overwhelm, and helplessness for parents caring for suicidal adolescents. While acknowledging the need for parents to assume diverse roles, we suggest that digital features in PiP+ should be developed to support parents to recognise and reflect upon their unique role and reach out to other supports as required (e.g., mental health professionals). As such, to adapt PiP+ there should be psycho-education content in the online modules about the role of each individual in the adolescent’s care team and content or activities to ensure the parent’s self-care which could prevent feelings of helplessness and overwhelm. Further, the program may also benefit from including a guided check-in between the coach and the parent about their self-care.

Lastly, the ‘holding hope for the future’ theme emphasises that to empower parents of suicidal adolescents with a digital intervention, there must be a belief that the adolescent will eventually recover and that the recommended parenting strategies will support their recovery. Consequently, PiP+ should be adapted to include digital features that emphasise parents’ progress across time and provide reflective mechanisms for parents to self-actualise their “gains” from engaging in the program. PiP+ could benefit from including a digital feature that allows parents to record and visualise their weekly learnings, progress, and strengths to foster greater parental empowerment. Such a record could be used collaboratively between parent and coach, as coaches can guide parents’ discoveries in ways to emphasise that recovery is possible. The findings of including self-narrative reflection are in line with educational research, which suggests that when students are asked to write a narrative where they can reflect upon themselves, their experiences, and learnings, this can support them to reframe what was previously perceived as difficult experiences ([9]). When students engage in self-narrative reflection, they no longer view perceived failures as setbacks, but rather opportunities for learning and growth ([9]).

The current study’s findings should be considered within the context of certain limitations. First, as the primary objective was to capture the breadth of perspectives about how parents can be more empowered when caring for a suicidal adolescent, we engaged with multiple stakeholder groups. A compromise, however, had to be made in terms of the smaller sample sizes within each stakeholder group, due to resource constraints. Thus, our findings may not be generalizable to the experiences and perspectives of all parents, young people, and experts. Second, expert demographic and professional details (e.g., age, therapeutic orientation) were not collected in the study as they were not considered central to the study aims; future studies may benefit from exploring on how such factors may shape experts’ perspectives. Third, the research team extrapolated the design features for the intervention by interpreting and extrapolating themes identified from prior workshops. Although these features were sense-checked by parents to confirm they align with the original themes, this approach carries a risk of confirmation bias—namely that the researcher’s interpretations of the participants’ responses may have shaped some of these features. Further, another limitation of this study is the potential for selection bias due to the recruitment of participants who had previously taken part in a related study ([8]) and had consented to future contact. These individuals may have had particularly strong motivations for participation, a higher level of engagement with research, or more positive or negative experiences that made them more likely to contribute again. As a result, the perspectives captured may not fully represent the broader population of parents, young people, or experts who have experience with adolescent suicidality but have not engaged in research. Future research should aim to build on such findings by engaging larger and more diverse groups to validate these proposed intervention features. Further, co-design methods could allow stakeholders to be able to generate multiple technological solutions as opposed to responding to the researcher-derived solutions.

## 5. Conclusions

In summary, the study examined how a digital intervention (PiP+) can be adapted to empower parents of adolescents experiencing suicidality, using co-designed digital features. The findings highlight the importance of empowering parents of suicidal adolescents by supporting parents to build suicide prevention skills and knowledge within their own parenting context, to have hope for the future, and to understand their role and their own needs. Specific technological features were identified that would foster such themes. To the best of our knowledge, there is currently no parenting intervention that has been co-designed to empower parents caring for adolescents experiencing suicidality. As such, these findings provide a preliminary step in considering how digital interventions could be developed to better support parents who are caring for an adolescent experiencing suicidality.

## Figures and Tables

**Figure 1 ejihpe-15-00199-f001:**
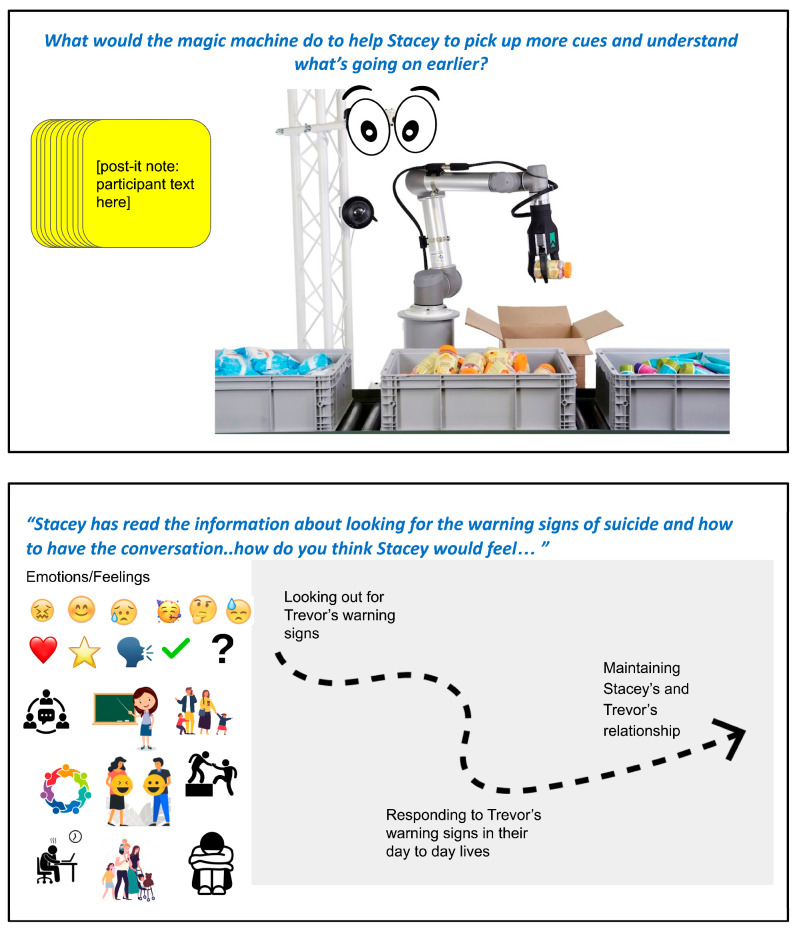
Initial parent co-design workshop activities. Top panel: magic machine for parental empowerment activity. Bottom panel: emotional enablers and barriers to empowerment activity.

**Figure 2 ejihpe-15-00199-f002:**
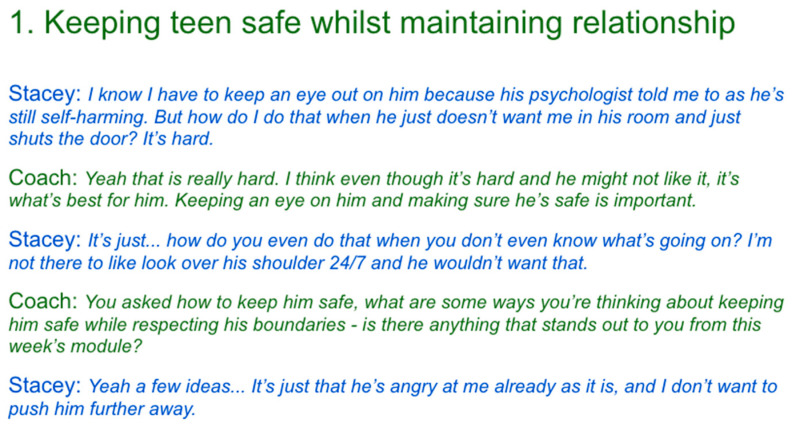
An exemplar vignette that experts were presented with in the co-design workshop.

**Figure 3 ejihpe-15-00199-f003:**
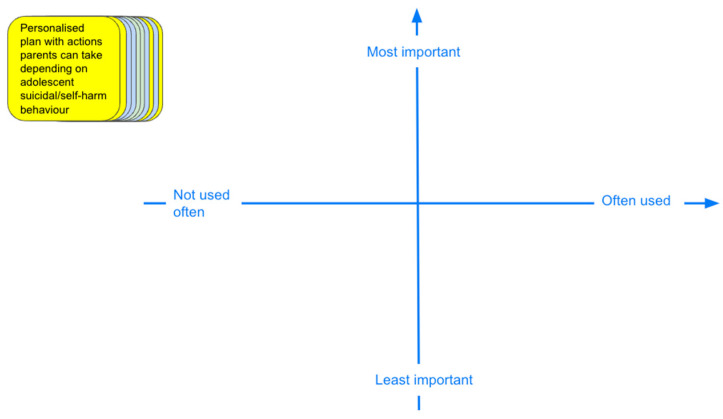
Parent sense-checking co-design workshop activity.

**Figure 4 ejihpe-15-00199-f004:**
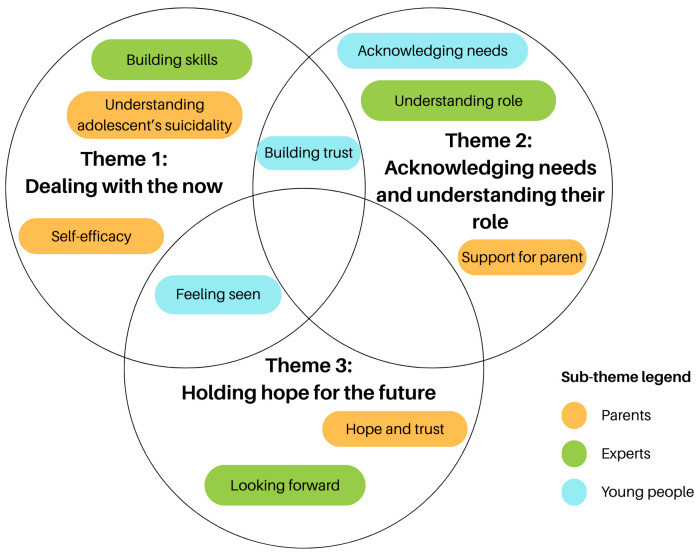
Thematic map of parental empowerment. The three themes of parental empowerment are shown in black: (1) *Dealing with the now*, (2) *Acknowledging needs and understanding their role*, and (3) *Holding hope for the future*. The ten sub-themes are represented as coloured nodes, with colours indicating the stakeholder group from which each sub-theme was derived.

**Figure 5 ejihpe-15-00199-f005:**
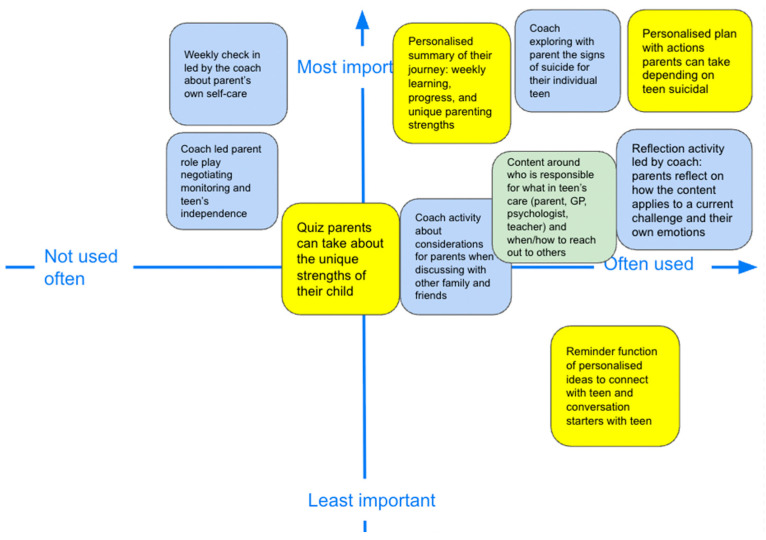
Parent prioritisation of proposed digital intervention features mapped along perceived importance and usability axes.

**Table 1 ejihpe-15-00199-t001:** Identified sub-themes from co-design workshops, the gap identified to be designed for, and suggested concretised feature in a digital, therapist-assisted parenting intervention to address the identified sub-theme.

Identified Sub-Themes	Gap Identified Which Needs to Be Designed for Parent Empowerment	Concretised Feature in a Digital, Therapist-Assisted Parenting Intervention to Address the Sub-Theme
Self-Efficacy	Feeling Prepared Awareness of Strengths	Personalised plan with actions parents can take depending on adolescent suicidal behaviour
Hope and Trust in Adolescent’s Recovery	Awareness of Progress	Personalised record of learnings: weekly learning, progress, and unique parenting strengths
Building Skills	Managing parent anxieties/distress tolerance Reflecting on how parenting strategies may be implemented in their own life	Reflection activity led by coach: parents reflect on how the content applies to a current challenge and their own emotions
Building Trust	Supporting the adolescent’s autonomy while monitoring for suicide risk	Coach-led parent role play of a situation that requires negotiation of monitoring and adolescent’s independence
Understanding Role	Parents’ understanding of their unique role in adolescent suicide prevention	Module content around who is responsible for what in adolescent’s care team (parent, GP, psychologist, teacher) and when/how to reach out to others
Support for Parent	How parents can seek support for themselves or their family in the context of the adolescent’s suicidality	Content and activity about considerations for parents when discussing with other family and friends
Understanding Adolescent’s Suicidality	Contextualizing suicidality for own adolescent	Coach exploring with parent the signs of suicide for their adolescent

**Table 2 ejihpe-15-00199-t002:** Demographics of stakeholder groups.

Demographic Characteristics	Parents	Young People	Experts
N	4	4	4
Gender			
Woman	4	2	3
Man	0	2	1
Mean Age ± SD	47.0 ± 4.94	21.0 ± 1	
Experts’ Professions ^a^			
Clinician Researcher			4

*Note.* ^a^ Clinical backgrounds included three psychologists and a psychotherapist.

**Table 3 ejihpe-15-00199-t003:** Identified sub-themes from co-design workshops, and indicative verbatim quotes of sub-themes.

	Description of the Sub-Theme	Indicative Verbatim Quotes of Sub-Themes
Dealing with the now sub-themes		
Self-efficacy	For parents to be able to acquire and draw upon specific skills and knowledge while their adolescent is suicidal, parents described that it was important for them to have a sense of confidence that they can support their adolescent. Parents described that without such confidence, it would be challenging for them to be able to draw upon specific suicide prevention skills and knowledge.	“[The magic machine] would give Stacey [the parent in the vignette] the confidence to just broach the subject with Trevor [the adolescent in the vignette]. It would give her all the vocabulary to use when she’s having discussions with him… it’d give her confidence to implement what she’s learned” (Parent 4).
Understanding their adolescent’s suicidality	Parents described that for them to be able to engage in skills and knowledge related to adolescent suicide prevention when their adolescent is experiencing a suicidal crisis, they need to understand why their adolescent is suicidal, and what are the signs of escalating suicidal thoughts and behaviours specific to their own adolescent.	“(I’d like to) understand what’s underlying that behaviour. So, if they’re staying in their room what’s underlying that, is it because they’re feeling terrible about themselves and don’t want to talk to anyone and going down a spiral of negative thinking or are they just enjoying their time.” (Parent 1)
Building skills	Experts described that for parents to be able to support their adolescent during a suicidal crisis, they need to be supported to develop skills to respond in line with evidence-based best-practice suicide prevention strategies.	“It should be quite clear that this is the goal of our work. This is not psychotherapy. This is coaching, which is focused on developing [the parents’] skills” (Expert 2)
Building trust	Young people described that for parents to be able to successfully implement suicide prevention strategies, there first needs to be a sense of trust between the parent and adolescent. They highlighted the importance that parents can be understanding of what their adolescent is experiencing, noting that without this sense of being understood, they were less likely to trust their parent’s advice, even when well-intentioned. As such, building trust in their parent’s emotional insights is essential for young people to be more receptive of accepting help during their suicidal crises.	“I don’t think I trusted her to know really what was going on. And so then I did not trust any of her advice. If you don’t have a relationship where like, you’re both like trusting each other, then it’s really hard to then trust things you’re telling me to do.” (Young person 3)
Feeling seen	Young people described that for parents to be able to successfully implement suicide prevention strategies, and for their adolescents to be more receptive to such strategies, parents need to understand their adolescents’ difficulties in the context of who they are and their current experiences. Young people indicated that they would be less open to seeking help and receiving support from their parents if they considered that their parents viewed them as being ‘broken’ or in need of ‘fixing’.	“(Parents should be) seeing (their adolescent) like a person and, and like understanding they have feelings… it can’t just be like, ’I want to fix my broken son’… (the adolescents) are going to like be like, ‘oh, so you think there’s like something wrong with me. Great. Like, wow, news flash, there is something wrong with me’… But if the intention is like, ‘oh, you know, I really want to support you through this so I understand what you’re going through’. Um, they’ll feel that too.” (Young Person 2)
Acknowledging needs and understanding their role sub-themes		
Support for parent	Parents described that to feel empowered to care for a suicidal adolescent, they need to be able to draw upon professional and personal support for themselves. Parents discussed such needs for support given that their consistent caretaking role can take a toll on their well-being.	“They’re still a child that needs us to be strong, which we can’t do all the time. And so we also need to have our own support and friends to help us when we’re struggling with all of that ourselves”. (Parent 1)
Understanding role	Experts described how parents need to be supported and to understand their role in caring for their adolescents to help them feel empowered. They discussed that at times, parents can feel that they are solely responsible for their adolescent’s health, education, and well-being. Therefore, parents need to understand that they are part of a team and can leverage the appropriate expertise of the adolescent’s care team (e.g., health professionals) to provide support.	“Their role is around support and being there rather than taking the control completely… For the parents to feel like… it’s not just them. It’s who else? Who else can the young person talk to? Who else can they get support from? So, it’s more of a shared team approach”. (Expert 1)
Acknowledging needs	Young people described the importance of parents reflecting and acting upon their own needs as a form of parent empowerment when caring for a suicidal adolescent. Further, young people described that when parents acknowledge their own needs and self-care, they also in turn model this skill to their adolescent.	“[I’d like her to] like self-care, like, ‘Oh, Mum’s actually taking care of herself. That’s cool. Maybe I should do a little bit myself or something’… Even modelling that behaviour is so important.” (Young person 2)
Building trust	Young people described how parents must trust the young person’s care team to uphold their duty of care in order to feel empowered in their role of supporting their adolescent.	“I was already connected with other supports that I was like, well, I’m already talking to all these people and I already am trying to work on all these things. You don’t have to be breathing down my neck to like work out what’s wrong or like work out if something has changed. Like someone else will work that out, you don’t need to work that out.” (Young person 3)
Holding hope for the future sub-themes		
Hope and trust	Parents described that to feel a sense of empowerment when engaging in a parenting intervention, parents need to hold hope that with time their adolescent will eventually recover and have trust that the parenting strategies they learn will support their recovery.	“[Parents need to] understand that it’s not necessarily going to change overnight… but you still have to keep trying and you still have to keep giving that support… you have to sort of remain hopeful.” (Parent 4)
Feeling seen and looking forward	Young individuals described the importance of parents recognising the current state and difficulties faced by adolescents (i.e., feeling seen). Yet, they equally emphasised that parents should maintain hope that recovery is possible although it may require time and continued effort (i.e., looking forward). Echoing young people’s descriptions, experts discussed that for parents to feel empowered, they need to believe that their adolescent’s suicidal thoughts and behaviours can be overcome by the parent supporting the adolescent holistically (i.e., not only focusing on suicidality).	“It’s like having enough faith that you’ll make it to the end but being realistic with where you’re at… Of course, like what [the adolescent] is going through is tough and is really, really like worrying. I’d be really worried. But this is, this is the long haul. And then also helping to understand that after you’re able to have a… having opened up, that’s not the end. Like there’s more to it.” (Young Person 2) “This is still your child. This is still, this person is going through a difficult time with all these things going on. So how do we keep seeing them and how do we keep hearing and holding out for who they are and the hope for the future and not get too fixated on the safety on its own.” (Expert 1).

## Data Availability

A summary of the data that support the findings of this study are available on request from the corresponding author, M.B.H.Y. The data are not publicly available owing to their containing information that could compromise the privacy of research participants.

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
