# Peer review of "Empowering Parents of Adolescents at Elevated Risk of Suicide: Co-Designing an Adaptation to a Coach-Assisted, Digital Parenting Intervention"

_ejihpe, 2025, doi:10.3390/ejihpe15100199_

Round 1

Reviewer 1 Report

Comments and Suggestions for Authors

Empowering Parents of Adolescents at Elevated Risk of Suicide: Co-designing an Adaptation to a Coach-Assisted, Digital Parenting Intervention

by

Alice Cao , Ling Wu , Glenn Melvin , Mairead Cardamone-Breen , Grace Broomfield , Joshua Seguin , Chloe Salvaris , Jue Xie , Dhruv Basur , Tom Bartindale , Roisin McNaney , Patrick Olivier , Marie Bee Hui Yap *

Thank you very much for this clear and engaging article on such an important issue.

The article highlights the problem that parents are the primary points of contact for adolescents at risk of suicide. The PiP+ digital program aims to support this group of parents and should be enhanced with the concept of parental empowerment. To achieve this, the methodology of co-designing is utilized to participatively outline the program's additions. The results demonstrate the success of this methodology, and the authors carefully address the existing limitations.

The article is well-organized and includes all necessary components for easy comprehension. Overall, the literature is well summarized, and the co-designing methodology is clearly explained. The relevance to the topic at hand is effectively demonstrated, making it logical to supplement the existing PiP and PiP+ programs with parental empowerment.

As an interested reader, I would appreciate a more comprehensive overview of other existing evaluated programs in the introduction, along with a critical examination of the PiP+ program. Additionally, the necessity of adding "parental empowerment" could be presented more substantively—perhaps through systematic reviews or meta-analyses instead of relying solely on individual primary studies.

Your presentation of the methodology is clear and easy to follow, as is the presentation of the results for various thematic areas of supplementation. The detailed explanation of the methodology is particularly interesting.

However, the presentation of the results on pages 10-13 could be condensed and presented in a tabular format, as the summary and interpretation that follow sufficiently reflect the results.

Author Response

Author Response:
Thank you for taking the time to review our manuscript and for your suggestions regarding its improvement, we really appreciate it. Please see below for a point-by-point response to your comments. 

Thank you very much for this clear and engaging article on such an important issue.

The article highlights the problem that parents are the primary points of contact for adolescents at risk of suicide. The PiP+ digital program aims to support this group of parents and should be enhanced with the concept of parental empowerment. To achieve this, the methodology of co-designing is utilized to participatively outline the program's additions. The results demonstrate the success of this methodology, and the authors carefully address the existing limitations.

The article is well-organized and includes all necessary components for easy comprehension. Overall, the literature is well summarized, and the co-designing methodology is clearly explained. The relevance to the topic at hand is effectively demonstrated, making it logical to supplement the existing PiP and PiP+ programs with parental empowerment.

As an interested reader, I would appreciate a more comprehensive overview of other existing evaluated programs in the introduction, along with a critical examination of the PiP+ program. Additionally, the necessity of adding "parental empowerment" could be presented more substantively—perhaps through systematic reviews or meta-analyses instead of relying solely on individual primary studies.

Your presentation of the methodology is clear and easy to follow, as is the presentation of the results for various thematic areas of supplementation. The detailed explanation of the methodology is particularly interesting.

However, the presentation of the results on pages 10-13 could be condensed and presented in a tabular format, as the summary and interpretation that follow sufficiently reflect the results.

Author Response
Thank you for the encouraging response regarding our manuscript. We appreciate your well-considered review. As suggested, the sub-theme descriptions and indicative verbatim quotes have now been presented in a tabular format (Table 3, p. 11). 

Reviewer 2 Report

Comments and Suggestions for Authors

The study seeks to co-design an adaptation of PiP+ to empower parents caring for suicidal adolescents, incorporating insights from parents, youth, and professionals. It explores how empowerment can be integrated into both the online modules and the digitally-mediated coaching components of the PiP+ system.

The manuscript is clear and relevant for the field addresses a gap in knowledge

Specific comments: Researchers should consider having participants' statements in quotes

Author Response

Author Response:
Thank you for taking the time to review our manuscript and for your suggestions regarding its improvement, we really appreciate it. Please see below for a point-by-point response to your comments.
The study seeks to co-design an adaptation of PiP+ to empower parents caring for suicidal adolescents, incorporating insights from parents, youth, and professionals. It explores how empowerment can be integrated into both the online modules and the digitally-mediated coaching components of the PiP+ system.

The manuscript is clear and relevant for the field addresses a gap in knowledge

Specific comments: Researchers should consider having participants' statements in quotes

Author Response
Thank you for your encouraging comments on our manuscript. We appreciate the time it has taken you to review our manuscript. As per your suggestion, we have added participants’ statements in quotation marks. As suggested by another reviewer, the sub-theme indicative verbatim quotes are now presented in a tabular format (Table 3, p. 10)

Reviewer 3 Report

Comments and Suggestions for Authors
  • (4-178-179) In the introduction, ‘Parents’  are mentioned, but as the sample description  indicates that there are only mothers, therefore from there, only ‘mothers’ should here be mentioned.
  • (4-157) In the text it is said: “Three groups were invited” ; it  may refer  to ‘experts’ , but not  to ‘fathers’ and ‘children’.  Moreover, the selection criteria employed should also be presented with more detail.
  • It is necessary to add more  data related to  experts’ age,  training and psychological orientation.
  • 4- (4-81) In addition to 2.3.2, it is said that there was an individualized activity with  children, but the authors  do not explain if it was different from the  work done with mothers and experts.  This should be indicated.
  • (7-270) At this point, in passing, mothers’ age are included. Such  data should appear  in the groups description, also adding  the demographic data of the other groups: young people and 

6- In the presentation of results  more concrete examples of the work meetings   content  should be included, that would permit to  readers understand  the process carried out by the IA.

Comments on the Quality of English Language

I have not the competences in this field

Author Response

Author Response:
Thank you for taking the time to review our manuscript, we greatly appreciate your thoughtful suggestions to improve our work. Please see below for a point-by-point response to your comments and concerns.

“(4-178-179) In the introduction, ‘Parents’  are mentioned, but as the sample description  indicates that there are only mothers, therefore from there, only ‘mothers’ should here be mentioned.”

Author Response:
We acknowledge that our sample in this study comprised only mothers, which is clarified in the methods section. 

“The final participants of this group co-design workshop were mothers aged 39 to 52” p. 7

However, we have intentionally retained the use of the broader term parents in the introduction, as this reflects both the wider literature on parenting interventions and the intended scope of the program, which is designed to support mothers and fathers alike.

(4-157) In the text it is said: “Three groups were invited” ; it  may refer  to ‘experts’ , but not  to ‘fathers’ and ‘children’.  Moreover, the selection criteria employed should also be presented with more detail. It is necessary to add more  data related to  experts’ age,  training and psychological orientation.

Author Response:
Thank you for this helpful comment. We have clarified in the text that the “three groups” invited to participate were (1) parents, (2) adolescents, and (3) professional experts, rather than implying that this referred only to experts. Further, we have addressed the description of participant criteria. Specifically: all participants needed to live in Australia, speak English, and have stable internet access. Parent participants were eligible if they had lived experience of parenting an adolescent (aged 12–18) who had experienced suicidal thoughts or behaviours. Young people (aged 15–25) were eligible if they had lived experience of suicidal thoughts or behaviours during adolescence (aged 12–18). Experts were eligible if they had more than 3 years of experience in youth mental health and suicide prevention, either in clinical or research roles. (p. 4)

Author Response:
We recognise that expert demographic and professional characteristics (e.g., age, training, orientation) may provide additional context. However, these data were not collected as they were not central to the study aims. We have clarified this in the manuscript and noted that future work could examine how such factors may shape experts’ perspectives.

“Second, expert demographic and professional details (e.g., age, therapeutic orientation) were not collected in the study as they were not considered central to the study aims; future studies may benefit from exploring how such factors may shape experts’ perspectives.” (p. 15) 

4- (4-81) In addition to 2.3.2, it is said that there was an individualized activity with  children, but the authors  do not explain if it was different from the  work done with mothers and experts.  This should be indicated.

Author Response: 

We have clarified in the manuscript that the activity for young people differed from those undertaken with parents and experts. Specifically, while parents and experts reflected more directly on parenting practices and intervention content, young people were asked to complete a vignette-based activity framed as providing advice from a position of empowerment.

(7-270) At this point, in passing, mothers’ age are included. Such  data should appear  in the groups description, also adding  the demographic data of the other groups: young people 

Author Response:
Thank you, the mothers’ ages are included in text as opposed to in the table, as it describes that 

“The final participants of this group co-design workshop were mothers aged 39 to 52 (M = 47.00, SD = 5.72).” p.17

The ages of the other participant groups (i.e., young people and experts) are listed in Table 1. 

6- In the presentation of results  more concrete examples of the work meetings   content  should be included, that would permit to  readers understand  the process carried out by the IA.

Author Response:
Thank you for this helpful comment, we have included some further detail about the meetings, and how digital features were extrapolated from sub-themes.

“Further, the research team had extrapolated digital design solutions that may address the needs identified by themes developed from parents, young people, and experts in the prior workshops. The lead author brainstormed potential features which could align with the sub-theme, and the concept of these features were then iteratively refined through multidisciplinary team discussions including human-computer interaction researchers, psychology researchers, and psychologists regarding how a feature could align with the needs and preferences articulated by stakeholder groups” p.7

Reviewer 4 Report

Comments and Suggestions for Authors

Introduction:

Although the topic is interesting, the introduction is overly lengthy, convoluted, and difficult to follow. There is no explanation for the literature gap, nor is there a clear rationale for conducting this research. Some paragraphs need to be rewritten and reorganized for clarity. For instance, Lines 113-131 present the intervention, which should be included in the methods section. Despite reading the paragraph multiple times, the reader remains confused about the different levels of intervention.

At the end of the introduction, the manuscript should outline clear objectives; however, the aims of this study are not presented concisely.  On the contrary, the authors are presenting the expected outcomes of the study.

Furthermore, the manuscript combines informal and formal writing styles, which makes it harder to follow. For example, the authors frequently use the phrase “that is” throughout the text. It is highly recommended that the authors choose a consistent writing style and adhere to it. The authors state “The findings will explore how empowerment could be embedded in the technological system (i.e., PiP+, which incorporates both computer and human  elements)—that is, how parent empowerment could be embedded in both the online mod-145 ules and digitally-mediated coaching sessions.”

Methods: There are major flaws in this section.

The study design and framework are either not clearly presented or are buried within a convoluted four-page methodology that makes them difficult to locate. Is the reader expected to assume that this study is either prospective or retrospective? Additionally, was this study conducted using a mixed-methods approach?

The methods section also includes results intertwined with the methodology. Certain elements, such as Tables 1 and 2, as well as Figures 1-3, should be moved to the results section.

Furthermore, the methods section must include all necessary information for another researcher to replicate the procedures. As it stands, the information provided is insufficient for replication.

How was the data analysis conducted? While the authors attempt to describe the coding process, they do not provide any thematic analysis.

Results: This section contains significant flaws.

The authors present Figure 4, leading readers to assume that the colors and labels might substitute for the missing interpretation. It is unclear whether "Looking Forward" or "Self-Efficacy" are intended to be sub-themes or main themes. It is highly recommended that each theme and sub-theme be clearly defined, and Figure 4 should be used to enhance the interpretation. Furthermore, the quotations should support the themes rather than dilute their meaning.

Discussion: There are significant flaws in this section.

For example, the paper begins by addressing the barriers and facilitators to parental control; however, these factors are not mentioned in the results section. This creates a notable disconnect between the results and the discussion.

The next paragraph (lines 489-508) focuses on the authors' interpretation of the results rather than providing a comprehensive discussion and interpretation of the findings. While subsequent paragraphs attempt to analyze the findings, they are largely ineffective. Additionally, the study should recommend areas for future research.

Conclusion: Since the results section does not present the findings in a logical manner, the conclusion lacks substantiation.

Comments on the Quality of English Language

The findings will explore how empowerment could be embedded in the technological system (i.e., PiP+, which incorporates both computer and human  elements)—that is, how parent empowerment could be embedded in both the online mod-145 ules and digitally-mediated coaching sessions.”

Author Response

Author Response:
Thank you for taking the time to review our manuscript and for your suggestions. Please see below for a point-by-point response to your comments. 

Introduction:

Although the topic is interesting, the introduction is overly lengthy, convoluted, and difficult to follow. There is no explanation for the literature gap, nor is there a clear rationale for conducting this research. Some paragraphs need to be rewritten and reorganized for clarity. For instance, Lines 113-131 present the intervention, which should be included in the methods section. Despite reading the paragraph multiple times, the reader remains confused about the different levels of intervention.

Author response:
We thank the reviewer for their feedback. Where possible we have reduced the word count of the introduction and re-organised the paragraphs to improve clarity. We respectfully disagree that we did not explain the literature gap which was outlined throughout the introduction, for example:

“While the literature consistently highlights calls for greater parental support, specifically for parents of suicidal adolescents (Calear et al., 2016; Lantto et al., 2023; Rheinberger, Shand, et al., 2023), very little is known about parents’ support needs when caring for a suicidal adolescent”  p.2
and,
“Parent empowerment was conceptualised as the belief that they have a valuable role in preventing their adolescent’s suicide and undertaking the behaviours to do so (Cao et al., 2025). Further, the field of human-computer interaction has highlighted empowerment as a key factor in designing effective interventions (Schneider et al., 2018). Yet, despite research calling for digital interventions to ‘empower’ its end users, the term ‘empowerment’ remains elusive and challenging to characterise and quantify (Schneider et al., 2018).” p. 2
and,
“Given the overlap in parental factors associated with adolescent internalising disorders and suicidality, a parenting intervention that addresses both appears as a promising path forward. Currently, there is no purely digital parenting intervention that exists for parents of suicidal adolescents, although a digital intervention does exist for parents of adolescents with internalising disorders that has pre-established acceptability, feasibility, and usefulness (Fulgoni et al., 2019)” p. 3

As such, we believe the literature gaps that we outlined lead to the aim of the study, as presented in page 3:

“The study aims to identify themes which would facilitate the empowerment of parents caring for a suicidal adolescent. In addition, the study will use co-design methodology to explore how the themes of empowerment could be embedded within the technological system (i.e., PiP+) through both self-directed online modules and digitally-mediated coaching sessions.” p.3

We agree that the section regarding the level of intervention was convoluted and as such, we have simplified the paragraph referencing in text the original paper should the reader wish to have greater context. 

“One such parenting intervention is the Partners in Parenting (PiP) program, which aims to reduce the risk and impact of adolescent depression and anxiety disorders by equipping parents with evidence-based parenting strategies. The Partners in Parenting program is a multi-level, individually tailored, web-based parenting intervention, spanning universal prevention (Level 1) to early intervention for adolescent internalising problems (Level 4) (see Yap et al., 2017 for more details). The highest level (Level 4; henceforth referred to as PiP+) combines the online self-directed program with regular one-on-one coaching sessions via video conference (Fulgoni et al., 2019). PiP+ involves up to nine interactive online modules covering different domains of evidence-based parenting (Fulgoni et al., 2019), and up to 13 video-conferencing sessions with their therapist-coach, occurring weekly or fortnightly (Fulgoni et al., 2019).”  p.3

At the end of the introduction, the manuscript should outline clear objectives; however, the aims of this study are not presented concisely.  On the contrary, the authors are presenting the expected outcomes of the study.

Author response:
We have rewritten this paragraph to more clearly outline the objectives of the study.

“The study aims to identify themes which would facilitate the empowerment of parents caring for a suicidal adolescent. In addition, the study will use co-design methodology to explore how the themes of empowerment could be embedded within the technological system (i.e., PiP+) through both self-directed online modules and digitally-mediated coaching sessions.” p 3

Furthermore, the manuscript combines informal and formal writing styles, which makes it harder to follow. For example, the authors frequently use the phrase “that is” throughout the text. It is highly recommended that the authors choose a consistent writing style and adhere to it. The authors state “The findings will explore how empowerment could be embedded in the technological system (i.e., PiP+, which incorporates both computer and human  elements)—that is, how parent empowerment could be embedded in both the online modules and digitally-mediated coaching sessions.”

Author response:
We appreciate the reviewer’s feedback regarding consistency of writing style, which is in line with feedback from Reviewer 5. We have reviewed our manuscript carefully to be more academic in tone throughout. 

 While we appreciate the comment regarding our use of ‘that is’, after carefully reviewing the manuscript, we found that the phrase “that is” appears twice, once in the section cited by the reviewer and in the sentence “Three groups of participants were invited to co-design workshops that is, (1) parents, (2) young people, and (3) experts in youth mental health/suicide prevention.” Nevertheless, we have addressed the reviewers' comments regarding our use of ‘that is”. Further, where possible, we have also reduced the use of ‘i.e.,’ throughout the manuscript. 

“The study aims to identify themes which would facilitate the empowerment of parents caring for a suicidal adolescent. In addition, the study will use co-design methodology to explore how the themes of empowerment could be embedded within the technological system (i.e., PiP+) through both self-directed online modules and digitally-mediated coaching sessions.” p.3

Methods:
There are major flaws in this section.

The study design and framework are either not clearly presented or are buried within a convoluted four-page methodology that makes them difficult to locate. Is the reader expected to assume that this study is either prospective or retrospective? Additionally, was this study conducted using a mixed-methods approach?

Author response:

Following from the reviewer’s feedback, we have added a study design section which outlines our methodology section at the outset. 

“This qualitative study used a co-design approach to adapt an existing digital parenting intervention (PiP+) for parents of adolescents experiencing suicidality (PiP-SP+). We adopted an inductive approach to identify themes which would facilitate the empowerment of parents caring for a suicidal adolescent, and then a deductive approach to validate whether the digital features suggested from such themes would support parents to feel empowered when caring for their suicidal adolescent. Three stakeholder groups were engaged: (1) parents, (2) young people, and (3) experts in youth mental health/suicide prevention, across four sets of online workshops. The first set of workshops was with parents, and was designed to elicit their perspectives on the enablers and barriers to empowerment. The second set of workshops was with young people, and was designed to capture their views on the acceptability of parent empowerment strategies. The third set of workshops, conducted with professional experts, was designed to gather their input regarding the feasibility of adapting a therapist-assisted online parenting program (PiP+), exploring systemic factors to empowerment. The final set of workshops was a sense-checking workshop with parents, designed to validate and refine themes and digital intervention features with parents “ pp. 3-4

The methods section also includes results intertwined with the methodology. Certain elements, such as Tables 1 and 2, as well as Figures 1-3, should be moved to the results section.

Author response:
Figures 1 to 3 represent exemplar activities that parents completed and as such are to be presented in the methodology. Further, Table 1 describes the participants’ demographics. While we acknowledge that Table 2 could be presented in results, given that it was generated directly from the co-design process and used within the final workshop to sense-check findings, we consider it methodologically integral and have retained it in the methods section.

Furthermore, the methods section must include all necessary information for another researcher to replicate the procedures. As it stands, the information provided is insufficient for replication.

How was the data analysis conducted? While the authors attempt to describe the coding process, they do not provide any thematic analysis.

Author response:
We thank the reviewer for raising this point. To clarify, our analytic approach was not thematic analysis but affinity mapping, which is a widely used method in co-design research to synthesise qualitative data. We apologise if this was not sufficiently clear in the manuscript. In our study, transcripts from all workshops were reviewed and corrected, then analysed collaboratively by two authors (AC and GB) using affinity mapping to display and cluster data from each stakeholder group. Sub-themes were inductively generated from these clusters and then iteratively refined into three overarching themes that represented perspectives across parents, young people, and experts. These procedures are described in Section 2.5 (Data Analysis), and we have revised this section to make the analytic process more explicit, including the steps of clustering, developing sub-themes and the thematic maps across the groups of stakeholders:

“Data was analysed using affinity mapping, a co-design analytic technique which involves visually clustering qualitative data to identify patterns and relationships (Baxter et al., 2024; Hanington & Martin, 2019; Parsell et al., 2024; Pernice, 2018). The first author AC and author GB familiarised and read through the transcripts and distilled participant insights into key points which were reduced to post-it notes in preparation for affinity mapping. Using the process of affinity mapping, concepts that were considered related to each other were grouped by AC and GB to form candidate sub-themes for each stakeholder group. AC and GB collaboratively integrated data across all workshops to identify candidate themes and sub-themes for empowering parents in supporting a suicidal adolescent in acceptable and feasible ways. The candidate sub-theme clusters were then grouped and further refined into overarching themes. Together, the overarching candidate themes and sub-themes were organised into a thematic map. All candidate themes and sub-themes were presented to the research team and iteratively discussed until the final themes and sub-themes were agreed upon.” p. 9

Results:
This section contains significant flaws.

The authors present Figure 4, leading readers to assume that the colors and labels might substitute for the missing interpretation. It is unclear whether "Looking Forward" or "Self-Efficacy" are intended to be sub-themes or main themes. It is highly recommended that each theme and sub-theme be clearly defined, and Figure 4 should be used to enhance the interpretation. Furthermore, the quotations should support the themes rather than dilute their meaning.

Author response:
We agree that some contextualisation of Figure 4 could help audience members interpret the figure. As such, we have included the following text to support readers’ interpretation of the figure.

“The thematic map (Figure 4) illustrates the overarching themes of parent empowerment when emotionally supporting their adolescent experiencing suicidality (in black text). Sub-themes identified by each participant group are indicated using a colour legend, orange for parents, blue for young people, and green for experts.” p. 9

In response to the comment regarding how each theme and sub-theme should be clearly defined, we had initially written the descriptions of the sub-theme in text, however, we appreciate that, given the number of sub-themes this could present challenges with readers following the description. As such, we have created a table with descriptions of sub-themes and indicative verbatim quotes of each sub-theme, which we believe would help readers align the description of the sub-theme and the indicative verbatim quotes. 

“Three sub-themes of ‘holding hope for the future’ were identified and are described with illustrative quotes in Table 3.” p. 11

Discussion:
There are significant flaws in this section.

For example, the paper begins by addressing the barriers and facilitators to parental control; however, these factors are not mentioned in the results section. This creates a notable disconnect between the results and the discussion.

Author response:
While we appreciate the reviewer’s comments, we respectfully disagree with the reviewer’s comment regarding a disconnect between the results and discussion. The paper does not discuss “barriers and facilitators to parental control”; rather, it presents enablers to parental empowerment. These concepts were central to the design of the workshops and are explicitly reported in the results section through the three overarching themes (“Dealing with the now,” “Acknowledging needs and understanding their role,” and “Holding hope for the future”) and their associated sub-themes. As such, the discussion provides a more in-depth interpretation of the overarching themes. We believe that the discussion builds directly on these results by interpreting how they align with or extend existing literature and by considering their implications for adapting PiP+. We wonder if the reviewer may have misinterpreted the terminology, as the focus throughout is on empowerment rather than control, and the barriers/enablers highlighted in the discussion are drawn directly from the results. 

The next paragraph (lines 489-508) focuses on the authors' interpretation of the results rather than providing a comprehensive discussion and interpretation of the findings. While subsequent paragraphs attempt to analyze the findings, they are largely ineffective. Additionally, the study should recommend areas for future research.

Author response:
While we appreciate the reviewer’s feedback regarding the discussion section. We note a possible inconsistency in the comment, in which the reviewer stated that the section (lines 489-508) focuses on interpretation rather than comprehensive discussion yet also suggests that the interpretation of findings is lacking. Our intention of the discussion was to provide an overview of the theme for empowerment, then situate such findings in the parenting intervention literature, and extend such findings to features. We also highlight that the discussion concludes with clear recommendations for future research, see below: 

“Future research should aim to build on such findings by engaging larger and more diverse groups to validate these proposed intervention features. Further, co-design methods could allow stakeholders to be able to generate multiple technological solutions as opposed to responding to the researcher-derived solutions.” p.16

As such, we believe the structure of our discussion provides both interpretation and a comprehensive discussion aligning with the journal’s expectations.

Conclusion: Since the results section does not present the findings in a logical manner, the conclusion lacks substantiation.

Author response:
While we thank the reviewer for their feedback, we note that this issue was not raised by the other four reviewers, all of whom found the structure and presentation of the results to be clear. We believe that the results section was organised to align with the study findings, and that the findings therefore flow logically. 

Comments on the Quality of English Language

The findings will explore how empowerment could be embedded in the technological system (i.e., PiP+, which incorporates both computer and human  elements)—that is, how parent empowerment could be embedded in both the online modules and digitally-mediated coaching sessions.”

Author response:
We have revised the quoted sentence as follows:

“The study aims to identify themes which would facilitate the empowerment of parents caring for a suicidal adolescent. In addition, the study will use co-design methodology to explore how the themes of empowerment could be embedded within the technological system (i.e., PiP+) through both self-directed online modules and digitally-mediated coaching sessions.” p.3

Reviewer 5 Report

Comments and Suggestions for Authors

Suicidal ideation and behaviours are common among adolescents, posing significant challenges. Parents play a fundamental protective role in the prevention of adolescent suicide. As such, it is important that parents feel empowered in this role, i.e., have greater self-efficacy and feel prepared to take the appropriate actions towards preventing adolescent suicide. An online parenting program could offer parents flexible access to evidence-based parenting strategies. Yet very little is known about how an intervention could be designed to support the empowerment of these parents.

Based on this the AUTHORS explore how to co-design an adaptation to an existing evidence-based, digital parenting intervention to empower parents. Four sets of co-design workshops with parents who have lived experience of caring for a suicidal adolescent (n = 4), young people who experienced suicidality during adolescence (n = 4), and experts in youth mental health/suicide prevention (n = 4) were conducted to innovate adaptations to an existing, evidence-based digital parenting program to empower parents of suicidal adolescents. Inductive thematic analysis was used to analyse and interpret findings.

Following these themes, the AUTHORS extrapolated ideas for technological features, which were presented to parents with lived experience (n = 3) to obtain their feedback. Three key themes highlight how a digital intervention could be innovated and adapted to empower parents caring for a suicidal adolescent.

That is, for parents to feel empowered to parent a suicidal adolescent, a digital intervention should support them to: 1) “deal with the now”; 2) “acknowledge needs and understand their role”, and 3) “hold hope for the future”. Further, ten sub-themes were developed illustrating different concepts related to these themes. Technological features were proposed by the AUTHORS to formulate how an intervention (Partners in Parenting - Suicide Prevention; PiP-SP+) could support parents to feel more empowered when caring for a suicidal adolescent.

THEIR proposed adaptations designed for PiP-SP+ offer a novel resource for parents and insights for clinicians and intervention designers, demonstrating how a digital intervention can be adapted to empower parents in their role of emotionally supporting and managing the suicide risk of their adolescent.

The study is truly very interesting.

These are my comments on how to improve it.

- The abstract has a very narrative and un-academic tone; with minimal effort, both the structure and the exposition could be improved.

- The methods could be improved in structure. Too much emphasis is placed on ethical authorization, which could be moved to the back matter, while the study design is somewhat vague. In fact, many paragraphs deal with statistics, sampling, data analysis, data collection, and recruitment, and there isn't a single paragraph that effectively presents the study. Include it along with a short summary that precedes the sections.

- The figures (ALL), starting with the methods section (where they are attractive figures), should be accompanied by a detailed description in the body of the text, otherwise they are not useful.

- The results are presented with sections that have eccentric titles (e.g., building with the now; acknowledging needs and understanding role; holding hope for the future...). The content is original and works, but try to provide a structure that's consistent with the goals.

- In your conclusions, start with a more academic presentation.

Author Response

Author Response:
Thank you for taking the time to review our manuscript and for your suggestions regarding its improvement, we really appreciate it. Please see below for a point-by-point response to your comments. 

Suicidal ideation and behaviours are common among adolescents, posing significant challenges. Parents play a fundamental protective role in the prevention of adolescent suicide. As such, it is important that parents feel empowered in this role, i.e., have greater self-efficacy and feel prepared to take the appropriate actions towards preventing adolescent suicide. An online parenting program could offer parents flexible access to evidence-based parenting strategies. Yet very little is known about how an intervention could be designed to support the empowerment of these parents.

Based on this the AUTHORS explore how to co-design an adaptation to an existing evidence-based, digital parenting intervention to empower parents. Four sets of co-design workshops with parents who have lived experience of caring for a suicidal adolescent (n = 4), young people who experienced suicidality during adolescence (n = 4), and experts in youth mental health/suicide prevention (n = 4) were conducted to innovate adaptations to an existing, evidence-based digital parenting program to empower parents of suicidal adolescents. Inductive thematic analysis was used to analyse and interpret findings.

Following these themes, the AUTHORS extrapolated ideas for technological features, which were presented to parents with lived experience (n = 3) to obtain their feedback. Three key themes highlight how a digital intervention could be innovated and adapted to empower parents caring for a suicidal adolescent.

That is, for parents to feel empowered to parent a suicidal adolescent, a digital intervention should support them to: 1) “deal with the now”; 2) “acknowledge needs and understand their role”, and 3) “hold hope for the future”. Further, ten sub-themes were developed illustrating different concepts related to these themes. Technological features were proposed by the AUTHORS to formulate how an intervention (Partners in Parenting - Suicide Prevention; PiP-SP+) could support parents to feel more empowered when caring for a suicidal adolescent.

THEIR proposed adaptations designed for PiP-SP+ offer a novel resource for parents and insights for clinicians and intervention designers, demonstrating how a digital intervention can be adapted to empower parents in their role of emotionally supporting and managing the suicide risk of their adolescent.

The study is truly very interesting.

Author Response:
Thank you for your enthusiasm for our manuscript and for your comments on how to improve our paper. We greatly appreciate the time you have put into reviewing our manuscript.

These are my comments on how to improve it.

- The abstract has a very narrative and un-academic tone; with minimal effort, both the structure and the exposition could be improved.

Author Response:
We agree with this suggestion, and as such, have rewritten the abstract to be more academic and structured in tone, as below:

“Suicidal ideation and behaviours are common among adolescents. Parents play a fundamental protective role in the prevention of adolescent suicide, but many describe feeling ill-equipped in their caretaking role. This is despite prior research indicating that it is important for these parents to feel empowered to emotionally support their adolescent if they are experiencing suicidality. An online parenting program could offer parents flexible access to evidence-based parenting strategies. However, there are limited digital resources for these parents and further, very little is known about how an intervention could be designed to support the empowerment of these parents. Therefore, the aim of the current study is to explore how an existing evidence-based, digital parenting intervention, Partners in Parenting (PiP+), could be adapted through co-design to empower parents. Four parents who have lived experience of caring for a suicidal adolescent, four young people who experienced suicidality during adolescence, and four experts in youth mental health/suicide prevention participated in four sets of co-design workshops to innovate adaptations to PiP+ to empower parents of suicidal adolescents. Affinity mapping was used to analyse and interpret findings. Three key themes highlight how a digital intervention could be innovated and adapted to empower parents caring for a suicidal adolescent. Specifically, for parents to feel empowered to parent a suicidal adolescent, a digital intervention should support them to: 1) “deal with the now”; 2) “acknowledge needs and understand their role”, and 3) “hold hope for the future”. Further, ten sub-themes were developed illustrating different concepts related to these themes. Findings highlight how technological features could support parents to feel more empowered when caring for a suicidal adolescent. In conclusion, the proposed technological features illustrate how digital interventions can be adapted to empower parents in their role of emotionally supporting and managing the suicide risk of their adolescent.” p.1

- The methods could be improved in structure. Too much emphasis is placed on ethical authorization, which could be moved to the back matter, while the study design is somewhat vague. In fact, many paragraphs deal with statistics, sampling, data analysis, data collection, and recruitment, and there isn't a single paragraph that effectively presents the study. Include it along with a short summary that precedes the sections.

Author response:
We agree that there was a significant emphasis on ethical authorisation, as such we removed the following.

This decision was made to ensure participants felt safe to provide their views freely and to prevent any perceived power imbalances between stakeholder groups unduly biasing or limiting the information shared.“
And
“While young people under the age of 18 were eligible to participate, no participants under the age of 18 expressed interest. Therefore, parental consent was not required.”

We appreciate the reviewer’s comments regarding the study design and as such have added in a section under “study design”:

“This qualitative study used a co-design approach to adapt an existing digital parenting intervention (PiP+) for parents of adolescents experiencing suicidality (PiP-SP+). We adopted an inductive approach to identify themes which would facilitate the empowerment of parents caring for a suicidal adolescent, and then a deductive approach to validate whether the digital features suggested from such themes would support parents to feel empowered when caring for their suicidal adolescent. Three stakeholder groups were engaged: (1) parents, (2) young people, and (3) experts in youth mental health/suicide prevention, across four sets of online workshops. The first set of workshops was with parents, and was designed to elicit their perspectives on the enablers and barriers to empowerment. The second set of workshops was with young people, and was designed to capture their views on the acceptability of parent empowerment strategies. The third set of workshops, conducted with professional experts, was designed to gather their input regarding the feasibility of adapting a therapist-assisted online parenting program (PiP+), exploring systemic factors to empowerment. The final set of workshops was a sense-checking workshop with parents, designed to validate and refine themes and digital intervention features with parents.” pp. 3-4

- The figures (ALL), starting with the methods section (where they are attractive figures), should be accompanied by a detailed description in the body of the text, otherwise they are not useful.

Author Response:
Thank you for this helpful suggestion, we have now incorporated a description of each figure within the body of the text to support their interpretation.

Figure 1 presents exemplar slides from the first parent workshop, showing the magic machine exercise and the emotion-mapping task to explore parental enablers and barriers to empowerment.” p. 5
“Figure 2 presents an exemplar slide in the expert workshop, depicting a vignette of a parent-coach conversation for experts to discuss strategies which could help overcome systemic barriers.” p.6
“Figure 3 presents a slide from the final parent workshop, where parents sorted the digital features of a personalised action plan along axes of importance and usability.” p.7

- The results are presented with sections that have eccentric titles (e.g., building with the now; acknowledging needs and understanding role; holding hope for the future...). The content is original and works, but try to provide a structure that's consistent with the goals.

Author Response:
We agree and thank you for your suggestion. We have changed the headers of our results section to more closely align with the research aims of our study. The results (i.e., theme names such as ‘dealing with the now) have been retained as sub-headings.

“3.1 Identified themes facilitating the empowerment of parents caring for a suicidal adolescent” p.10
“3.1.1 Dealing with the now” p.10
“3.2 How empowerment could be embedded in the technological system” p.13

- In your conclusions, start with a more academic presentation.

Author Response:
We agree with this suggestion, and as such, have rewritten the Conclusion to be more academic in tone, as below:

“In summary, the study examined how a digital intervention (PiP+) can be adapted to empower parents of adolescents experiencing suicidality, using co-designed digital features. The findings highlight the importance of empowering parents of suicidal adolescents by supporting parents to build suicide prevention skills and knowledge within their own parenting context, to have hope for the future, and to understand their role and their own needs. Specific technological features were identified that would foster such themes. To the best of our knowledge, there is currently no parenting intervention that has been co-designed to empower parents caring for adolescents experiencing suicidality. As such, these findings provide a preliminary step in considering how digital interventions could be developed to better support parents who are caring for an adolescent experiencing suicidality.” p.16

Round 2

Reviewer 3 Report

Comments and Suggestions for Authors

The corrections have been taken into account

Author Response

Thank you for taking the time to review our manuscript and for your suggestions regarding its improvement. We really appreciate it. 

We agree with your review that the objectives of our study could be more clearly stated. Therefore, we have revised the objectives to improve clarity. 

The study aims to explore the empowerment of parents caring for a suicidal adolescent through a qualitative design. A secondary aim is to explore how parental empowerment can be embedded within a technological system (i.e., PiP+ through both self-directed online modules and digitally-mediated coaching sessions).”  p.3

Reviewer 4 Report

Comments and Suggestions for Authors

While the authors have addressed some of my suggestions, the manuscript still requires revision.

For instance, the objectives are unclear and would benefit from more qualitative terminology for the healthcare arena. A possible rephrasing could be: "This study aims to explore the empowerment of parents through a qualitative design. A secondary aim is to…". The authors have included excessive information in the aims, which is more appropriate for the study design section.

Methods: Although the manuscript states that this is a qualitative study, the first paragraph lacks references, which could be considered plagiarism.

As previously mentioned, Table 1 should be moved to the results section. Additionally, Table 2 also belongs in the results section, as the methods are presented in a convoluted manner that is difficult to follow.

Figure 4: The figure should label each theme, such as "Theme 1: Dealing with the Now." However, the authors have not provided an explanation for the figure. The purpose of creating a figure is to present a summary of the results. While it has improved, it still requires further information to enhance clarity.

Author Response

Author response: Thank you for taking the time to review our manuscript and for your suggestions regarding its improvement. Please see below for a point-by-point response to your comments in blue. 

While the authors have addressed some of my suggestions, the manuscript still requires revision.

For instance, the objectives are unclear and would benefit from more qualitative terminology for the healthcare arena. A possible rephrasing could be: "This study aims to explore the empowerment of parents through a qualitative design. A secondary aim is to…". The authors have included excessive information in the aims, which is more appropriate for the study design section.

 Author response: Thank you for your suggestion, we agree and we have revised the objectives to improve clarity. 

The study aims to explore the empowerment of parents caring for a suicidal adolescent through a qualitative design. A secondary aim is to explore how parental empowerment can be embedded within a technological system (i.e., PiP+ through both self-directed online modules and digitally-mediated coaching sessions).”  p.3

Methods: Although the manuscript states that this is a qualitative study, the first paragraph lacks references, which could be considered plagiarism.

 Author response:
We thank the reviewer for highlighting the importance of referencing in the methods section. We respectfully disagree that the paragraph constitutes plagiarism. The paragraph referenced (Methods, Section 2.1 Study Design) is an original account of the procedures undertaken in this study, written to communicate the overall co-design methodology, stakeholder groups, and sequence of workshops. As such, it is not drawn from any single source.
Further, our manuscript already includes references for specific methodological elements adapted from previous studies, for example:

“The magic machine activity encourages participants to be imaginative and limitless in their design suggestions (Almohamed et al., 2020; Andersen & Wakkary, 2019).” pp. 4-5
“Data was analysed using affinity mapping, a co-design analytic technique which involves visually clustering qualitative data to identify patterns and relationships (Baxter et al., 2024; Hanington & Martin, 2019; Parsell et al., 2024; Pernice, 2018).” p. 8

However, to further support readers, we have added additional references to publications to the first paragraph that outline similar co-design methodologies to provide readers with additional context in the literature.

“This qualitative study used a co-design approach (Slattery et al., 2020; Tomitsch et al., 2018) to adapt an existing digital parenting intervention (PiP+) for parents of adolescents experiencing suicidality (PiP-SP+; Yap et al., 2013)”. p.3

As previously mentioned, Table 1 should be moved to the results section. Additionally, Table 2 also belongs in the results section, as the methods are presented in a convoluted manner that is difficult to follow.

Author response: Thank you for your recommendation regarding the participant demographics being moved to the results section. We have moved this table to the results section. pp. 8-9

As acknowledged in our prior response to your comment regarding Table 2, while we recognise that Table 2 could be presented in results, given that it was generated directly from the co-design process and used within the final workshop to sense-check findings, we consider it methodologically integral and therefore, we have retained it in the methods section.

Figure 4: The figure should label each theme, such as "Theme 1: Dealing with the Now." However, the authors have not provided an explanation for the figure. The purpose of creating a figure is to present a summary of the results. While it has improved, it still requires further information to enhance clarity.

Author response: We agree with the reviewer’s suggestion, and as such, we have revised both the text, figure, and also figure description. pp 9 - 10. 

The thematic map (Figure 4) presents a summary of the overarching themes and sub-themes of parental empowerment when emotionally supporting their suicidal adolescent. Themes are presented in black text, whereas sub-themes identified by each participant group are indicated using a colour legend (orange for parents, blue for young people, and green for experts). Areas of overlap between sub-themes reflect conceptual connections across stakeholder perspectives, to illustrate both shared and unique contributions.”

"

Figure 4. Thematic map of parental empowerment. The three themes of parental empowerment are shown in black: (1) Dealing with the now, (2) Acknowledging needs and understanding their role, and (3) Holding hope for the future. The ten sub-themes are represented as coloured nodes, with colours indicating the stakeholder group from which each sub-theme was derived. Areas of overlap between sub-themes reflect conceptual connections across stakeholder perspectives, to illustrate both shared and unique contributions. “ pp. 9-10.

Round 3

Reviewer 4 Report

Comments and Suggestions for Authors

The authors addressed my suggestions.